# Cross-sectional health centre and community-based evaluation of the impact of pneumococcal and malaria vaccination on antibiotic prescription and usage, febrile illness and antimicrobial resistance in young children in Malawi: the IVAR study protocol

David Singleton [1], Ana Ibarz-Pavon,[1,2] Todd D Swarthout [3,4]
Farouck Bonomali,[2] Jennifer Cornick,[1,2] Akuzike Kalizang'oma,[2,4] Noah Ntiza,[2]
Comfort Brown,[2] Raphael Chipatala,[5] Wongani Nyangulu,[5] James Chirombo,[2]
Gift Kawalazira,[6] Henry Chibowa,[7] Charles Mwansambo,[8]
Kenneth Mphatso Maleta,[5] Neil French,[1] Robert S Heyderman [4]

DS and AI-P contributed equally. NF and RSH contributed equally.

For numbered affiliations see end of article.

**Correspondence to**
Dr David Singleton;
D.A.Singleton@liverpool.ac.uk

## ABSTRACT

**Introduction** Vaccination is a potentially critical component of efforts to arrest development and dissemination of antimicrobial resistance (AMR), though little is known about vaccination impact within low-income and middle-income countries. This study will evaluate the impact of vaccination on reducing carriage prevalence of resistant *Streptococcus pneumoniae* and extended spectrum beta-lactamase-producing *Escherichia coli* and *Klebsiella* species. We will leverage two large ongoing cluster-randomised vaccine evaluations in Malawi assessing; first, adding a booster dose to the 13-valent pneumococcal conjugate vaccine (PCV13) schedule, and second, introduction of the RTS,S/AS01 malaria vaccine.

**Methods and analysis** Six cross-sectional surveys will be implemented within primary healthcare centres (n=3000 users of outpatient facilities per survey) and their local communities (n=700 healthy children per survey): three surveys in Blantyre district (PCV13 component) and three surveys in Mangochi district (RTS,S/AS01 component). We will evaluate antibiotic prescription practices and AMR carriage in children ≤3 years. For the PCV13 component, surveys will be conducted 9, 18 and 33 months following a 3+0 to 2+1 schedule change. For the RTS,S/AS01 component, surveys will be conducted 32, 44 and 56 months post-RTS,S/AS01 introduction. Six health centres in each study component will be randomly selected for study inclusion. Between intervention arms, the primary outcome will be the difference in penicillin non-susceptibility prevalence among *S. pneumoniae* nasopharyngeal carriage isolates in healthy children. The study is powered to detect an absolute change of 13 percentage points (ie, 35% vs 22% penicillin non-susceptibility).

## STRENGTHS AND LIMITATIONS OF THIS STUDY

⇒ The study builds on two large-scale vaccine evaluations, leveraging the infrastructure, methodology and community engagement developed by such evaluations.

⇒ The study expands on a range of studies in Malawi evaluating antibiotic exposure and development of antibiotic resistant pathogen carriage, while also developing new methods using established methods as comparators.

⇒ The study will enable methodologies to be evaluated against two vaccine delivery scenarios: (1) adaptation of delivery schedule of a pre-existing vaccine (13-valent pneumococcal conjugate vaccine) and (2) introduction of a new vaccine (RTS,S/AS01).

⇒ Though our study design may limit representativeness, we have opted for a largely pragmatic design due to operational challenges in this setting.

⇒ Despite monitoring throughout the study, there is nevertheless a risk of contamination between intervention and non-intervention arms (ie, children receiving an RTS,S/AS01 vaccine who relocate to a zone where RTS,S/AS01 is not being introduced or vice versa).

**Ethics and dissemination** This study has been approved by the Kamuzu University of Health Sciences (Ref: P01-21-3249), University College London (Ref: 18331/002) and University of Liverpool (Ref: 9908) Research Ethics Committees. Parental/caregiver verbal or written informed consent will be obtained prior to inclusion or recruitment in the health centre-based and community-based activities, respectively. Results will be disseminated via the Malawi

Ministry of Health, WHO, peer-reviewed publications and conference presentations.

## INTRODUCTION

Antimicrobial resistance (AMR) is a leading global health threat,[1] with 1.27 million deaths being attributable to AMR in 2019 alone.[2] AMR development is thought to be primarily driven by antimicrobial exposure,[3] but with resistant genes and their host bacteria capable of passing between people, animals and the environment[4] multifaceted approaches are needed if we are to curb AMR development and dissemination.[4] In 2016, a global review on AMR set out 10 recommendations for tackling this global pandemic,[5] 1 of which was vaccination.[5 6]

Vaccines may directly and indirectly impact on AMR.[7] Directly, vaccines target bacterial species with emerging clinical resistance issues (eg, *Streptococcus pneumoniae, Salmonella typhi*).[8–10] However, vaccines that target viruses and parasites may also deliver an indirect effect on AMR.[7 11] This may be via (1) removal of a 'gateway' pathogen for bacterial infection, (2) improving general health or (3) reducing frequency of symptoms commonly associated with antibiotic prescription (eg, fever),[12] thereby reducing selection pressure for AMR development.[7 11] Conversely, vaccines may also exert a resistance selection pressure on target and/or 'bystander' pathogens.[13 14] Thus, for us to understand these complex interactions more fully, it is crucial that the putative impact of vaccines on AMR is evaluated in a systematic manner.

Varying responses to vaccination have been observed between high-income (HIC) and low-income and middle-income countries (LMICs), with vaccines frequently underperforming expectations in LMICs despite high vaccine coverage rates.[15] However, many LMICs also have severely limited access to diagnostics and appropriate antibiotics,[16] with empirical and potentially unnecessary antibiotic prescriptions being a common reality of clinical practice.[17] Hence, while there is an intrinsic need to conduct vaccine impact evaluations in both HICs and LMICs, there is also a need to evaluate whether vaccination can play a cost-effective[18] role in assisting equitable provision of antibiotics to those at greatest need.

Pneumococcal conjugate vaccines (PCVs) have arguably received the most attention of any vaccine regarding their potential to reduce AMR.[7] In the USA and the UK, the introduction of 7-valent (PCV7) and, later, 13-valent (PCV13) vaccines were associated with considerable reductions in resistant pneumococcal infections.[19–21] Malawi introduced PCV13 in November 2011 using a 3+0 delivery schedule (1 dose at 6, 10 and 14 weeks of age), with vaccine coverage exceeding 90%.[22] Though introduction was associated with significant reductions in invasive pneumococcal disease[23] and all-cause mortality,[24] vaccine serotype carriage has remained persistently high,[25] and penicillin non-susceptibility in both carriage[15 26] and disease samples[27] has not decreased, particularly in non-vaccine serotypes.[15] Hence, some have queried whether PCV13 delivery can be further optimised.[28] The two PCV13 delivery schedules recommended by WHO include a 3+0 and a 2+1 (1 dose at 6 and 14 weeks of age and a booster at 9 months of age) schedule.[29] Given that some countries reporting reductions in AMR following PCV13 introduction use a booster dose,[19–21] it is important to evaluate the role of a booster dose in reducing pneumococcal carriage, disease, antibiotic prescriptions and, ultimately, AMR. In 2021, a pragmatic, cluster-randomised evaluation on the impact on pneumococcal carriage of changing the existing 3+0 PCV13 delivery schedule to a 2+1 schedule was implemented in the Blantyre district of Malawi (known as the 'PAVE' study)[30]; our study will leverage the PAVE study to also assess the impact of delivery schedule change on AMR.

In 2021, RTS,S/AS01 became the first malaria vaccine to be recommended by the WHO for widespread use in young children.[31] A subunit vaccine targeting *Plasmodium falciparum*, it is delivered via three doses at 5, 6 and 7 months of age followed by a fourth dose at 18–21 months of age.[32] Phase III trials indicated that the RTS,S/AS01 vaccine was effective at reducing clinical malaria.[33] However, although rare, increases in febrile convulsions, meningitis, cerebral malaria and mortality rates in RTS,S/AS01 vaccinated individuals[34] led to a recommendation for further safety profiling and impact assessment.[32] Following this, the WHO announced phase IV evaluations in selected areas in Ghana, Kenya and Malawi.[32] We hypothesised that reductions in malaria-associated febrile illness in young children may be associated with reductions in antibiotic exposure and therefore AMR.[12] RTS,S/AS01 vaccination may also be associated with generalised improvements in health, reducing antibiotic exposure,[7 11 12] and this study will work in conjunction with the ongoing phase IV evaluations in the Mangochi district of Malawi[32] to assess these hypotheses.

Hence, the 'Impact of Vaccines on Antimicrobial Resistance' (IVAR) study aims to leverage the PAVE and phase IV RTS,S/AS01 evaluations to assess the impact of (1) changing from a 3+0 to a 2+1 PCV13 delivery schedule and (2) introducing a novel non-bacterial vaccine (RTS,S/AS01) on antibiotic prescription, febrile illness and AMR carriage in children ≤3 years of age in Malawi. The IVAR study will also deploy methods for measuring antibiotic prescription and exposure in primary healthcare and community settings. As such, this study addresses three key research questions:

1. Can vaccination reduce the prevalence of antibiotic-resistant pneumococcal and extended spectrum beta-lactamase-producing (ESBL) *Escherichia coli* and *Klebsiella* species carriage in healthy children?
2. Can vaccination reduce incidence of febrile illness?
3. Can vaccination reduce incidence of antibiotic prescription?

the population.[35] Throughout the district there are 28 governmental primary care health centres (HCs)[30] which serve defined catchment areas (figure 2). The Malawi Ministry of Health and Blantyre District Health Office (DHO) randomly selected 10 HCs to implement a WHO-approved 2+1 PCV13 delivery schedule (intervention arm), with 10 other HCs randomly selected to continue the 3+0 schedule and serve as the comparator arm[30]; the IVAR study will work with a random selection of these HCs.

The IVAR study, in collaboration with an ongoing RTS,S/AS01 phase IV evaluation (ClinicalTrials.gov NCT03806465; RTS,S/AS01 introduced April 2019), will also work to evaluate the impact of RTS,S/AS01 on AMR in Mangochi district, southern Malawi (figure 1). Mangochi district is predominantly rural and is composed of 1.1 million residents over 6700 km$^2$, with children under the age of 5 comprising 18% of the population.[35] Throughout the district there are 29 governmental primary care HCs[36] which serve defined catchment areas (figure 3). Within this setting, RTS,S/AS01-exposed (n=5 HCs) and non-exposed (n=3 HCs) clusters have already been selected as part of a separate RTS,S/AS01 introduction evaluation[32]; the IVAR study will work with a random selection of these HCs. Vaccine-exposed HCs were geographically adjacent to each other (ie, exposed HCs are in Mangochi township itself or within <13 miles of the town, whereas non-exposed HCs are 14–38 miles outside of Mangochi township).[32]

### Study site selection

For the PCV13 component of this study, we will randomly select 6 HCs for IVAR study inclusion, including n=3 HCs switching to 2+1 and n=3 HCs continuing to provide 3+0, stratified by setting (urban, periurban and rural HCs) (figure 2). These will be selected from the ten 2+1 and ten 3+0 HCs that have already been selected for inclusion in the aforementioned PCV13 schedule change evaluation. Of note, a 3+1 PCV13 schedule was implemented among a subset of children following the schedule change, targeting children living in the catchment area of a 2+1 HC and who had received their first or second PCV13 primary doses prior to the HC implementing the schedule change.[30] These children will be eligible for recruitment.

For the RTS,S/AS01 component of this study, we will randomly select 6 HCs for IVAR study inclusion (n=3 RTS,S/AS01-exposed and n=3 non-exposed HCs) from the five RTS,S/AS01 exposed and three non-exposed HCs,[32] stratified by setting (urban, peri-urban and rural HCs) (figure 3).

### Study design

There will be 6 HC-based and community-based cross-sectional surveys, split between the PCV13 (n=3 surveys) and the RTS,S/AS01 component (n=3 surveys) evaluations. For the PCV13 component, surveys will be conducted 9 months, 18 months and 33 months

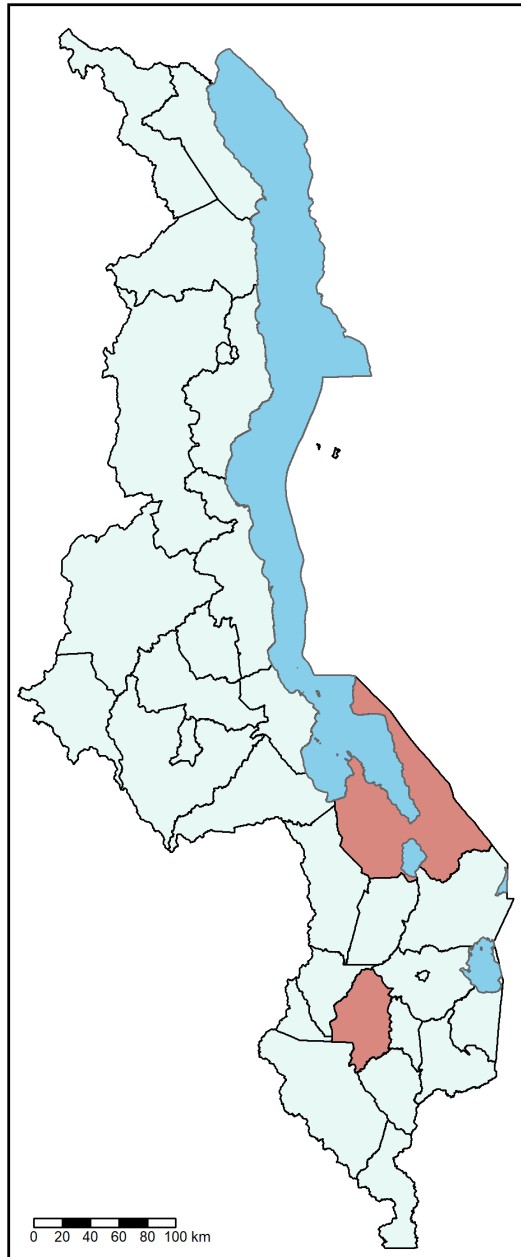

**Figure 1** Malawi split by 28 district and city regions. Red shading indicates the Mangochi (bordering Lake Malawi, shaded in blue) and Blantyre districts (central Southern Malawi) where the IVAR study will be conducted. IVAR, Impact of Vaccines on Antimicrobial Resistance.

## METHODS AND ANALYSIS
### Study setting

The IVAR study, in conjunction with the PAVE study[30] (ClinicalTrials.gov NCT04078997, implemented 16 March 2021), will evaluate the impact on AMR of changing the 3+0 to a 2+1 delivery schedule in children ≤3 years of age in the Blantyre District, southern Malawi (figure 1).[30] Blantyre district is a mixed urban and rural setting composed of 1.3 million residents over 1800 km$^2$, with children under the age of 5 comprising 16% of

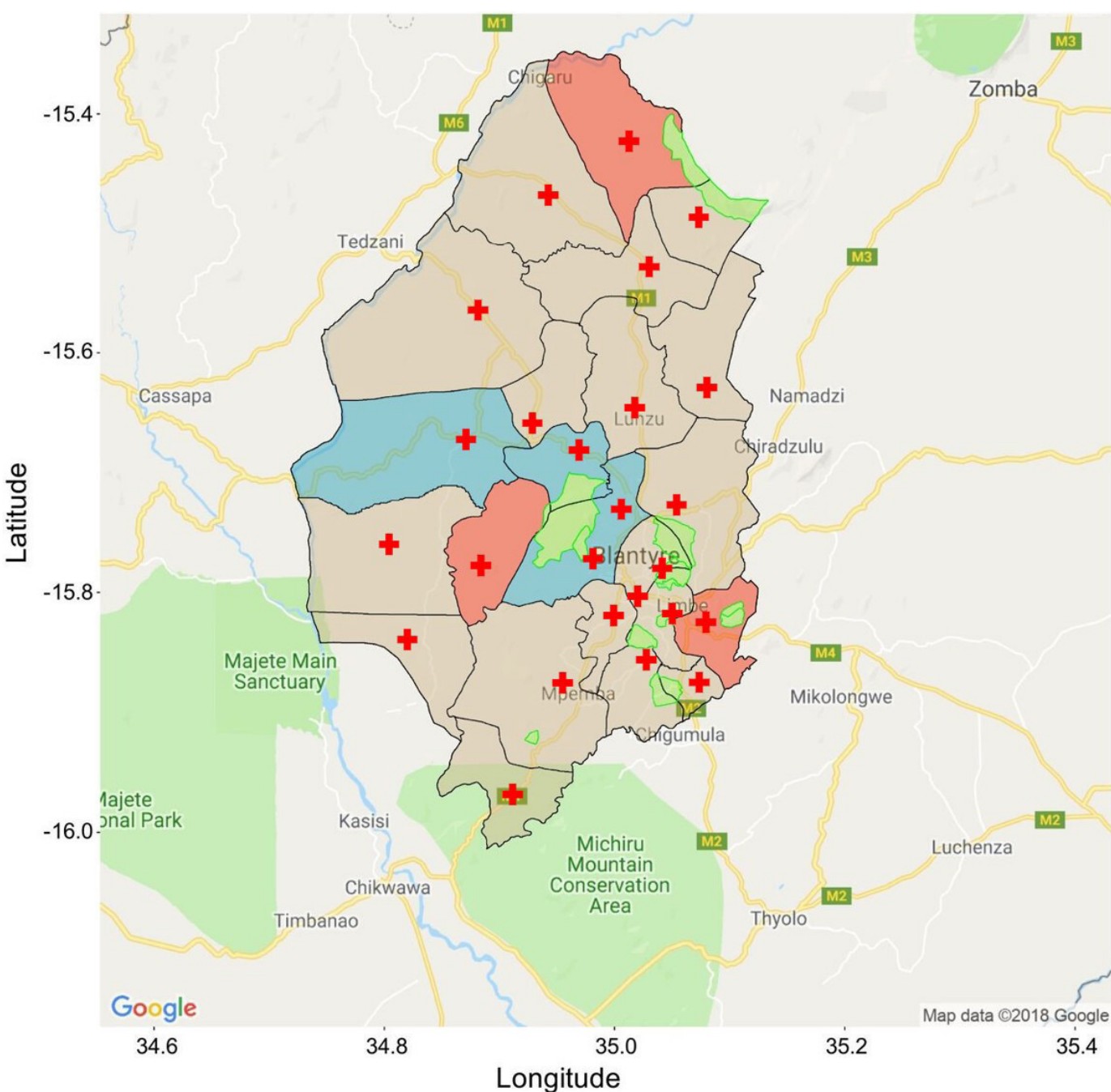

**Figure 2** Blantyre district with boundaries of 28 health centre catchment areas. Red cross=health centre. Blue and red shading indicate 2+1 and 3+0 PVC13 schedule areas randomly selected for this study, respectively. Green areas are non-inhabited (including mountains, industrial zones and other regions administratively declared not for habitation). Adapted from Swarthout *et al*.[30] PVC13, 13-valent pneumococcal conjugate vaccine.

post-schedule change implementation. For the RTS,S/AS01 component, surveys will be conducted 32, 44 and 56 months post-introduction. The surveys will be composed of two concurrent data collection activities. First, a community-based carriage survey of healthy children will be implemented, with collection of biological samples (nasopharyngeal and rectal swabs) and collection of information pertaining to the history of the child's febrile illness, malaria and antibiotic prescription history. Second, we will implement an anonymised audit of malaria rapid diagnostic test (mRDT) use and medicinal prescriptions (with a focus on antibiotics) in children presenting unwell to the outpatient department (OPD) of HCs, as recorded within each child's health passport (HP). While for all surveys the HC audit will summarise children ≤3 years of age, the age-based eligibility of the community carriage survey will vary according to survey and vaccine evaluation; a study sampling frame is included in figure 4. Data collection for this study was initiated in December 2021 and is scheduled to complete in March 2024.

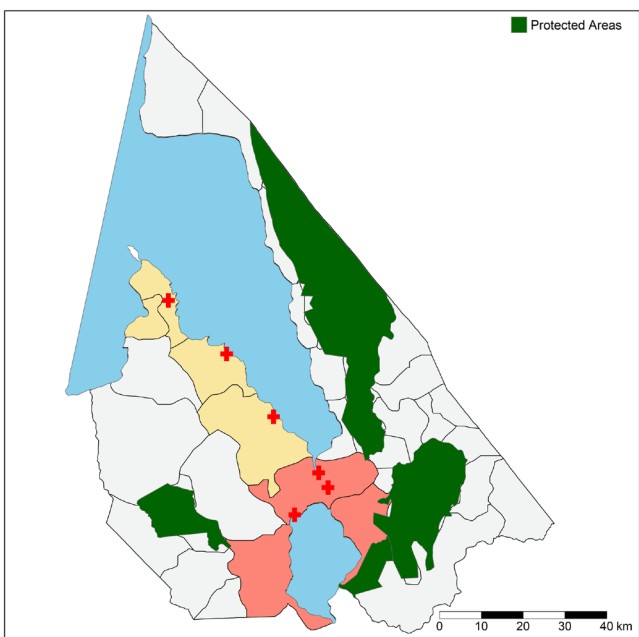

**Figure 3** Mangochi district with boundaries of 29 health centre catchment areas. Red cross=health centre. Red and yellow shading indicate RTS,S/AS01 malaria vaccine exposed and unexposed areas randomly selected for this study, respectively. Blue shading corresponds to Lake Malawi and Lake Malombe. Green areas are protected zones (eg, wildlife reserves and national parks).

## Primary objectives

The primary objective is to evaluate the reduction in carriage prevalence of penicillin resistant *S. pneumoniae* (see 'Community carriage survey: Inclusion and exclusion criteria' for further details), ESBL *E. coli* and *Klebsiella*

species following a PCV13 schedule change (Blantyre district) and, separately, following RTS,S/AS01 vaccine introduction (Mangochi district). This will answer the question of whether vaccines can play a role in reducing the carriage prevalence of resistant pathogens in young children in LMICs.

## Secondary objectives

The secondary objectives are to evaluate (A) incidence of febrile illness and antibiotic prescription and exposure (thus providing a mechanism for any reduction in AMR), (B) incidence of macrolide, tetracycline and trimethoprim-sulfamethoxazole non-susceptibility in *S. pneumoniae* and (C) changes to the wider upper respiratory tract and gastrointestinal resistome variation following PCV13 schedule change and RTS,S/AS01 vaccine introduction.

## Data collection activities
### HC audit
Data will be collected on vaccine cluster designation (ie, receiving intervention or not), vaccine schedule compliance, medicines prescribed and febrile illness presentation (using mRDT use as a proxy) during that visit via review of the child's HP. This audit is intended to assess the impact of PCV13 schedule change or RTS,S/AS01 introduction on mRDT use and antibiotic prescription frequency.

### *Population and sampling*
The audit will include children ≤3 years of age presenting to the OPD of selected HCs. Each HC will be assessed over at least 2 weeks during each survey; HCs in opposing study arms (ie, 2+1 vs 3+0; RTS,S/AS01 vs no RTS,S/AS01)

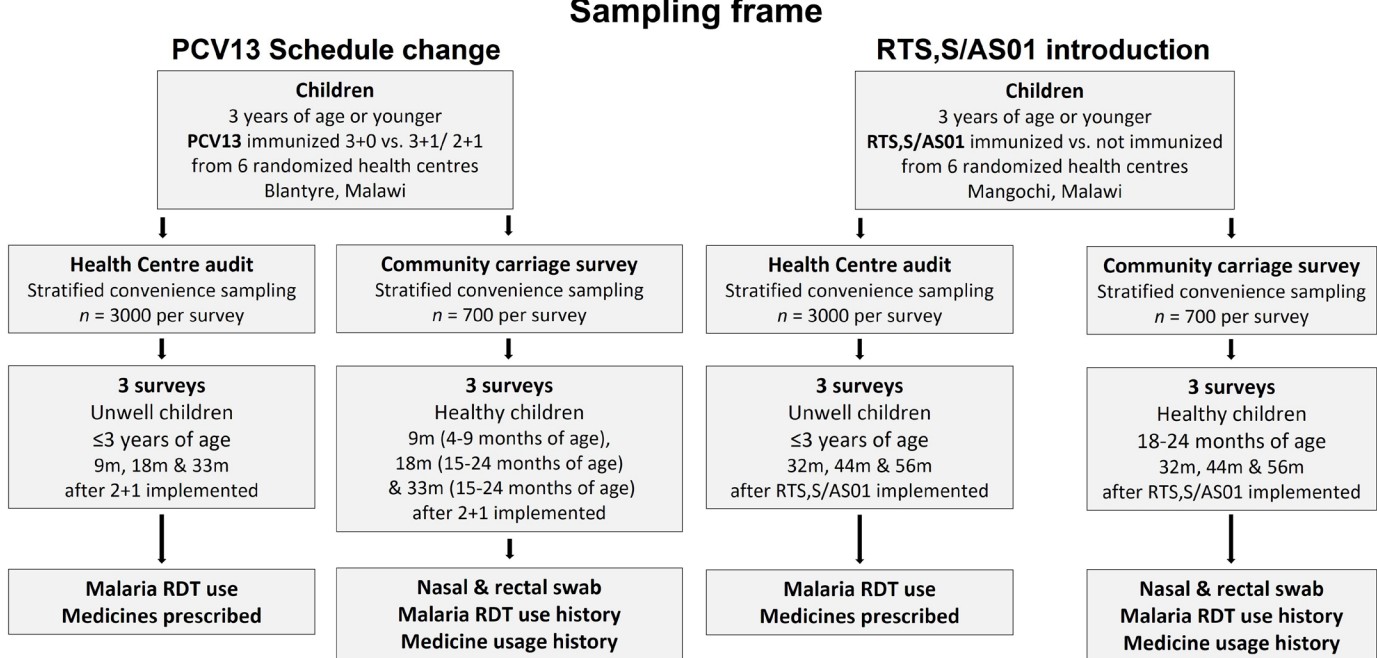

**Figure 4** IVAR study sampling frame. IVAR, Impact of Vaccines on Antimicrobial Resistance; PCV, pneumococcal conjugate vaccine; RDT, rapid diagnostic test.

will be paired according to setting and will be surveyed concurrently or on adjacent weeks depending on staff availability and local conditions (including weather patterns and related accessibility). Children's HPs will be reviewed on HC exit. Study teams will aim to review the HP of every eligible child; however, to avoid disrupting HC workflow, during busy periods study teams will not request that potentially eligible children wait until a member of the study team is available to review their HP.

### Inclusion and exclusion criteria
Inclusion criteria will be: (1) ≤3 years of age, (2) HC OPD attendance for investigation and/or treatment of ill health, (3) verbal consent has been provided by parent/caregiver for HP review and (4) HP available for review. Exclusion criteria will be: (1) HC attendance for vaccination and (2) HC attendance for routine health examination (eg, weighing) and found to be well. As HPs are reviewed anonymously and HPs are reviewed on a per visit basis, children may be recorded more than once within each survey.

### Intervention
Children's HPs will be reviewed on children exiting selected HCs, having presented to the OPD.

### Expected outcomes
The primary outcome will be the difference in antibiotic prescription incidence between intervention arms, measured as a proportion of total OPD visits of children ≤3 years of age within each surveyed period. A secondary outcome will be difference in mRDT use, again as a proportion of total OPD visits. In this study, we use mRDT as a proxy for febrile illness. Additional outcomes include subanalyses by specific antibiotic agent/class. Depending on availability of current census data, analyses may also encompass overall OPD visit incidence relative to paediatric populations within each health centre's catchment area.

### Study power and sample size calculation
A previous HC OPD review estimated that 70% of children under 5 years of age were prescribed antibiotics (Priyanka Patel, Malawi-Liverpool-Wellcome (MLW), personal communication). Hence, the primary outcome is powered to detect a difference of 5% in antibiotic prescription incidence between intervention groups (65% vs 70%). Two-tailed sample sizes were calculated setting confidence at 95% and power at 80%, indicating that 1417 HPs would be required in each study arm. Hence, we will review 3000 HPs per survey (n=18 000 for total study). HC-level sample sizes will be weighted according to estimated or actual population sizes within their respective catchment areas.[32]

### Informed consent process
Due to an anticipated high participation rate, we have developed a method, approved by the relevant research ethic committees, to maximise efficiency in attaining verbal consent and capturing the needed information. Parents/caregivers will receive verbal (and written if they wish) information about the study activity, and will have the opportunity to ask questions and express their doubts and concerns before accepting to take part. Verbal parent/caregiver consent will then be provided voluntarily if they choose to participate.

### Data collection, management and anonymisation procedures
Data from a questionnaire will be collected by password-protected electronic data capture (online supplemental file 1). Participation will be completely anonymised, with no personal data collected. Data will be uploaded daily to a secured on-site server, which is backed up daily to both local and off-site facilities.

### Statistical analyses
Only categorical variables will be collected; these will be defined by frequency distributions. Descriptive analyses encompassing both outcomes and vaccination status will be performed. Mixed effects logistic regression models investigating presence of (1) antibiotic prescription and (2) mRDT use on the day of HC attendance in individual children as outcome variables will be implemented. HC will be modelled as a random effect, with findings being balanced against the opposing outcome variable; vaccination status; other medicinal prescriptions; HC setting and month of HC visit. Individual antibiotic classes/agents may also be explored as subanalyses. Data collected here will be anonymously compared against routine attendance registry data collected by selected HCs.

## Community carriage survey
The community carriage surveys will focus on healthy children (figure 4). Nasopharyngeal and rectal swab samples will be collected in addition to demographic data, vaccine compliance, febrile illness, history of mRDT use and medicine prescription and exposure history. These data will be informed via direct parent/caregiver questioning, review of the child's HP, and an antibiotic provision recall exercise known as the 'drug bag method'.[16]

### Population and sampling
A stratified convenience sampling approach will be applied, making use of available local census data and utilising networks of Health Surveillance Assistants (HSAs) and health volunteers when needed. The sampling approach is intended to maximise efficiency in recruiting children who have received the appropriate intervention for the HC catchment area in which they reside. Hence, sampling will identify villages that are most proximal to HCs and will work to recruit all eligible children within those villages. Additional villages will be approached if needed until sampling targets are met.

## PCV13 component
Children aged 4–9 months will be sampled in the first survey to establish the baseline difference between children provided with three (3+0 schedule) or two (2+1

schedule) primary PCV13 doses. Children residing in 2+1 catchment areas and who have received three primary PCV13 doses (referred to as '3+1 children') will be eligible for study inclusion. In the second and third surveys, children 15–24 months of age will be sampled to compare children who have received a booster dose (2+1) against those who have not (3+0).

There are limited census data available in these settings. Thus, we will work closely with local HSAs and health volunteers who will assist with development of community engagement strategies (including communication with community leaders), identify eligible children, locate targeted households and facilitate communication with household members.

### RTS,S/AS01 component
Children aged 18–24 months of age will be sampled across all three surveys to compare children who have received the RTS,S/AS01 vaccine to unvaccinated children. Recent censuses have been completed to support the RTS,S/AS01 evaluation,[32] providing greater confidence in local population estimates. Hence, parents/caregivers of potentially eligible children will be located and contacted directly by study teams.

#### Inclusion and exclusion criteria
For both the PCV13 and RTS,S/AS01 components, inclusion criteria include (1) age of child within the range determined by the particular survey (figure 4), (2) permanent residence in the relevant study site, (3) parent/caregiver providing written informed consent, (4) evidence of having received a full initial (primary dose/s) vaccine course particular to evaluation and study arm and (5) that the child is healthy at time of sampling.

Exclusion criteria include (1) child having received antibiotics within the previous 14 days, (2) child currently receiving tuberculosis treatment, (3) child having been hospitalised for pneumonia within the previous 14 days, (4) presence of gross respiratory pathology, (5) child having a terminal illness, (6) child previously recruited into the current survey and (7) parent/caregiver not providing informed written consent.

For the second and third surveys of the PCV13 component, children residing in 2+1 clusters must have received the booster PCV13 dose. For the RTS,S/AS01 component, receipt of the booster dose is not a requirement in all three surveys, though provision will be recorded. Children may only be included in each survey once; however, if they fulfil the inclusion criteria they may be enrolled in subsequent surveys.

For both the PCV13 and RTS,S/AS01 components, sample collection will include nasopharyngeal and rectal swabs from each participant. Following previously described WHO recommendations,[25 37] nasopharyngeal swabs will be collected in skim milk-tryptone-glucose-glycerine medium and rectal swabs will be collected in Cary-Blair medium, both being stored in −80°C freezers at the MLW Research Programme laboratory in Blantyre

within 10 hours of collection for later batch-testing. Samples collected in Mangochi District will initially be taken to the Public Health and Nutrition Research Group laboratory, collocated to the Mangochi District Hospital, for storage prior to transport to MLW. Samples will be cultured to isolate and characterise *S. pneumoniae*, ESBL *E. coli* and *Klebsiella* species at MLW. Isolates will then be sent to the UK for whole genome sequencing (WGS).

For *S. pneumoniae*, penicillin non-susceptibility will be defined genotypically, as a minimum inhibitory concentration (MIC) over 0.06 mL/L; 200 genotypically non-susceptible isolates will be phenotypically tested to confirm genotypic findings. Non-susceptibility to macrolides (azithromycin, MIC 0.25 <mg/L), tetracyclines (doxycycline, MIC 1.00 <mg/ML) and trimethoprim-sulfamethoxazole (MIC 1.00 <mg/ML) will also be examined genotypically with phenotypic confirmation as secondary objectives. Subanalyses will include determining serotype of *S. pneumoniae isolates*. A total of 800 carriage samples will be selected for broader resistome analyses.

*E. coli* and *Klebsiella* species ESBL status will be determined phenotypically via chromogenic ESBL media. Depending on resource availability, a proportion of isolates will also undergo WGS and broader resistome analyses.

#### Data collection
A range of demographic, mRDT use and medicine exposure history will be collected from the participants' HP. These data will be supplemented with direct parent/caregiver questioning. The drug bag method will also be used; this has previously been deployed in Malawi[16] and we will repurpose this approach to assist parent/caregiver recall in identifying the different antibiotics given to the participating child. In brief, prior to study initiation the study team will obtain antibiotics routinely used for systemic administration (including multiple formulations of the same agent where obtainable) and available for dispensing or sale in the community (including HCs, hospitals, private pharmacies and informal sources). Parents/caregivers will be asked if they recognise individual antibiotics (presented as pictures, example in figure 5). For recognised antibiotics, parents/caregivers will be asked whether they have ever given the antibiotic to the participating child. If yes, they will be asked whether the antibiotic was given in the 12 months, 3 months or 14 days prior to recruitment (figure 6). Findings from direct questioning, HP review and the drug bag exercise will be compared and combined to provide a more complete history of a participant's antibiotic exposure.

Study questionnaires are available in online supplemental file 2 (PCV13 component, survey 1) (online supplemental file 3) (PCV13 component, surveys 2 and 3) and online supplemental file 4 (RTS,S/AS01 component, all surveys).

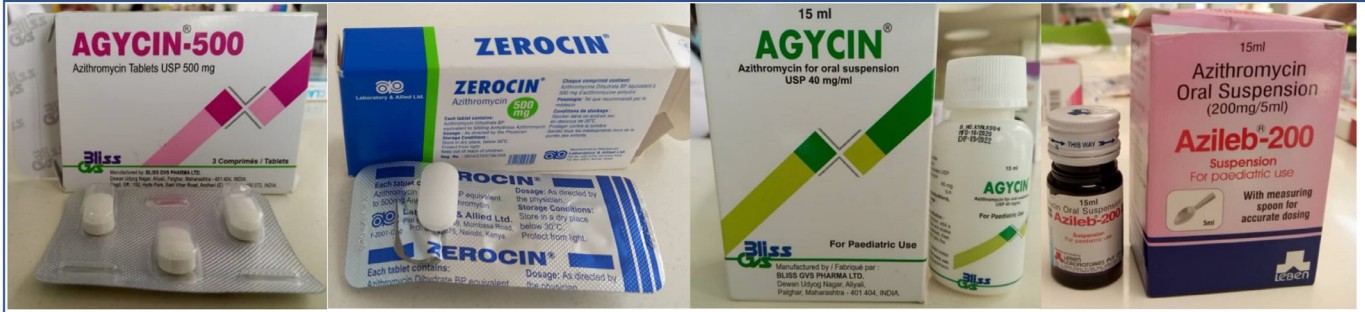

**4. AZITHROMYCIN**
AZM, agycin, zerocin, azileb

**Figure 5** Example of an antibiotic picture (azithromycin) which will be used in the drug bag exercise as part of the community carriage survey. Method adapted from Dixon *et al*.[16]

### Primary and secondary outcomes

#### Carriage isolates

The primary outcome will be the difference in prevalence of penicillin non-susceptibility among *S. pneumoniae* carriage isolates, comparing (1) the 2+1 vs 3+0 intervention arms of the PCV13 component) and (2) the RTS,S/AS01 vs no-RTS,S/AS01 intervention arms of the RTS,S/AS01 component. Secondary outcomes will also encompass macrolide, tetracycline and trimethoprim-sulfamethoxazole non-susceptibility.

#### Rectal swabs

The primary outcome will be the difference in prevalence of ESBL *E. coli* and *Klebsiella* species carriage comparing (1) the 2+1 vs 3+0 arms of the PCV13 component) and (2) the RTS,S/AS01 vs no-RTS,S/AS01 arms of the RTS,S/AS01 component.

#### Antibiotic usage

The primary outcome will be the difference in incidence of antibiotic prescription intended for systemic administration from 14 days to 3 months prior to recruitment, comparing (1) the 2+1 vs 3+0 arms of the PCV13 component) and (2) the RTS,S/AS01 vs no-RTS,S/AS01 arms of the RTS,S/AS01 component. Additional outcomes will include subanalyses by specific antibiotic agent/class and length of treatment.

#### Febrile illness

A secondary outcome will be the difference in incidence of febrile illness from 14 days to 3 months prior to recruitment, comparing (1) the 2+1 vs 3+0 arms of the PCV13 component) and (2) the RTS,S/AS01 vs no-RTS,S/AS01 arms of the RTS,S/AS01 component. Additional secondary outcomes will include subanalyses of mRDT use and results (including multiple positive tests per individual) and antimalarial treatment, comparing respective study arms.

### Study power and sample size calculation

Previous surveys have indicated a minimum pneumococcus carriage prevalence of 60% in children under the age of 5,[25] with 35% of such isolates expected to display penicillin non-susceptibility (unpublished data). Hence, the primary outcome is powered to detect a crude absolute decrease of 13 percentage points (13%) in penicillin non-susceptibility (ie, 35% vs 22%). Two-tailed sample sizes were calculated setting confidence at 95% and power at 80%, indicating that 204 pneumococcus isolates would be required in each study arm per survey. Hence, allowing for a 60% carriage prevalence, 700 samples will be collected per component per survey (n=4200 for total study). HC-level sample sizes will be weighted according to estimated or actual population sizes within their respective catchment areas.[32]

A previous survey has indicated a minimum ESBL *E. coli* carriage of 25% in Blantyre (Nicholas Feasey, MLW, personal communication). Hence, the sample size per survey noted above will allow a crude absolute difference of 8 percentage points (8%) to be detected (ie, 25% vs 17%) between study arms.

### Informed consent process

Participant's parents/caregivers will receive written and verbal information about the study, and will have the opportunity to ask questions, express their doubts and concerns, and have time to reflect before deciding to take part or not. An informed consent form will be signed and dated by the participant's parent/caregiver and a member of the research team, a copy of which will be retained by the parent/caregiver. Participant's parents/caregivers may withdraw consent at any point without need to provide a reason, and without penalty.

### Data collection, management and anonymisation procedures

Data will be collected using password-protected electronic data capturing. Each participant will be assigned a unique participant identification number (PID) at recruitment. This PID will be used in all datasheets and files, and will

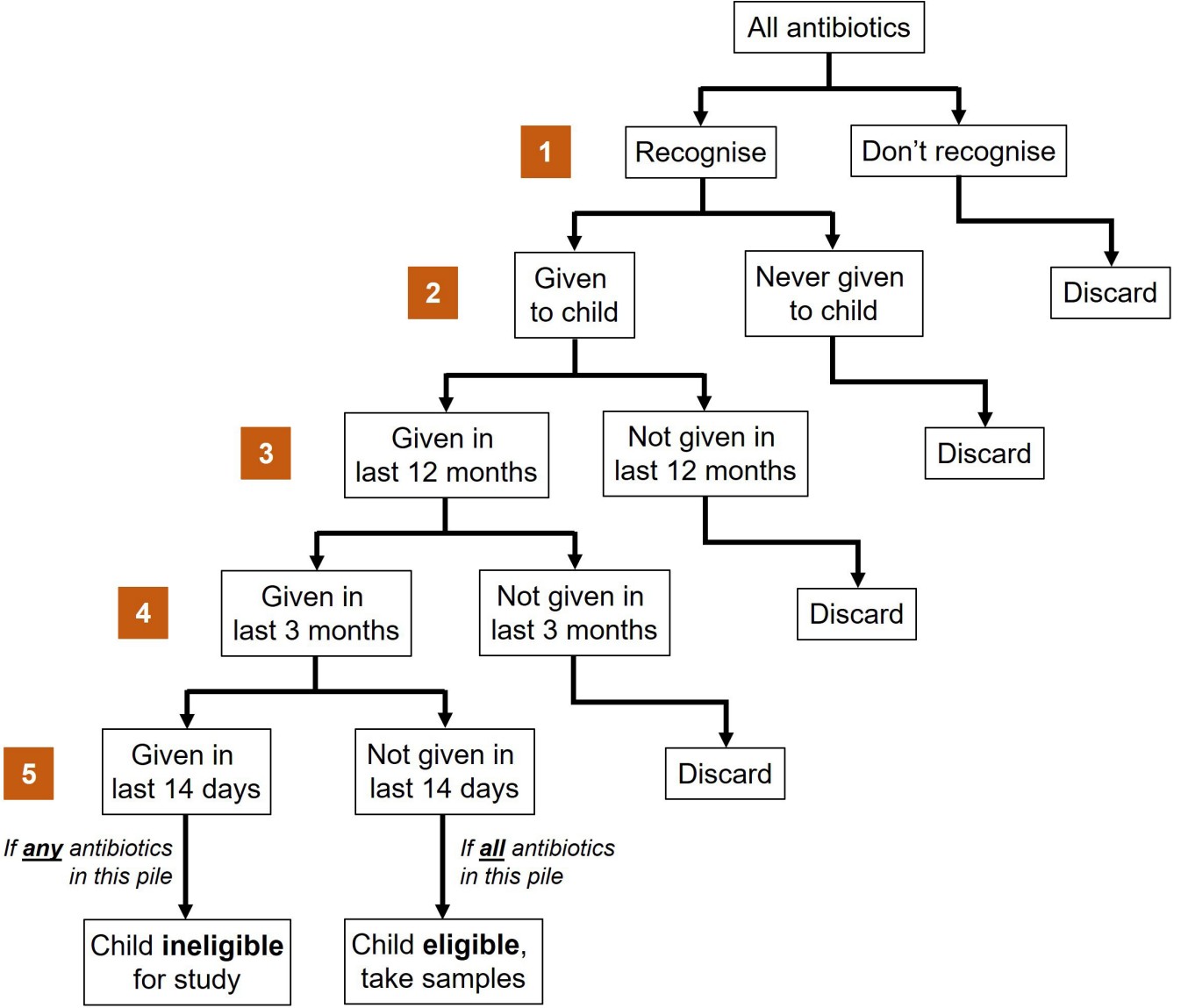

**Figure 6** Flow diagram of questionnaire workflow of the 'drug bag' component of the community carriage survey. Each orange number corresponds to a round of questioning, starting with all locally identified, available antibiotics being presented to participant's parents/caregivers. The pool of antibiotics available for review is expected to diminish in each successive round, with the final round (round 5) being included as a final check that no child has been given antibiotics within the 14 days prior to sampling, which would render them ineligible for survey participation.

be linked to laboratory data, thus, only anonymised data will be used for analyses. A logbook containing identifiable information (including name) will be kept separately in a secured location and will only be accessed by authorised study team members. This will allow the study team to recover any missing epidemiological information, if necessary, later (eg, missing vaccination dates), and to facilitate any participant's parents/caregivers who wish to withdraw consent.

### Statistical analyses

Continuous variables will be expressed as means and SD, or medians and IQRs. Categorical variables will be defined by frequency distributions. Descriptive analyses encompassing primary, secondary and additional outcomes, and vaccination status, will be performed. Mixed effects logistic regression models investigating prevalence of penicillin non-susceptible *S. pneumoniae* across all pneumococcus carriage isolates as an outcome variable will form the primary analysis. HC will be modelled as a random effect, with findings being balanced against key demographic variables (ie, age, sex), vaccination status, antibiotic and febrile illness history, rurality and the month in which sampling was completed. This approach will be repeated for other antibiotic classes, and for ESBL *E. coli* and *Klebsiella* species. We will also descriptively analyse antibiotic prescription and febrile illness history (using mRDT usage as a proxy), and may progress to inferential statistics if justified.

## Patient and public involvement

Prior to development of the protocol, key stakeholders were informed of the study, including the selected HCs and their surrounding communities, the DHO, Ministry of Health and the Ministry of Education. We actively sought and incorporated input from these stakeholders into the study objectives and overall design. Given that this study collaborates closely with two existing vaccine evaluations, pre-existing community engagement and sensitisation will strengthen community trust at the onset of this study. Especially in light of COVID-19, we envisage expanded community engagement prior to start of data collection, encompassing HCs, community leaders, HSAs, health volunteers and members of the communities themselves in information-providing activities. We will repeat community engagement activities prior to each survey and provide feedback on prior surveys where possible.

## Ethics and dissemination

### Ethical approval

This study has been approved by the Research Ethic Committees (REC) of Kamuzu University of Health Sciences (Ref: P01-21-3249), University College London (Ref: 18331/002) and the University of Liverpool (Ref: 9908) Research Ethics Committees. Parental/caregiver verbal (health centre audit) or written (household carriage survey) informed consent will be obtained prior to inclusion or participation, as described earlier.

### Dissemination policy and plans

Study results will be shared with local communities and stakeholders, the Malawi Ministry of Health, other relevant policy-makers and decision-making stakeholders, and published in peer-reviewed journals. Findings will be presented at international conferences and meetings. Copies of all published materials and reports will be shared with the research ethics committees and collaborators. Procedures for strain exchange, data sharing and ownership will follow Nagoya protocol standards.[38]

## DISCUSSION

To date, attempts to define an impact of vaccination on antimicrobial exposure and resistance patterns have largely been restricted to randomised controlled trials (RCTs) and postauthorisation retrospective analyses,[7 39 40] generally within HICs.[39] However, of note, Lewnard *et al* used LMIC household survey data collected 2006–2018 to demonstrate significant reductions in antibiotic exposure associated with introduction of childhood pneumococcal and rotavirus vaccines.[41] While promising, such impacts may be reduced in the longer term, particularly in high carriage prevalence settings[42] or via serotype replacement.[43] Indeed, following PCV13 introduction in Malawi, high residual pneumococcal carriage has been observed 8 years post-PCV13 introduction,[25] along with emergence of resistant serotypes.[15] Thus, there is a need to conduct a thorough evaluation in a high carriage, low-income setting.

Here, we have leveraged two existing evaluations,[30 32] seeking to define impact following vaccine intervention in Malawi. Though this has enabled more efficient study preparation, it does mean that we are dependent on existing evaluation methodologies. For example, both evaluations use a cluster randomised approach. This is a reasonable approach to take in a country where population censuses are infrequent and where individual, blinded randomisation would prove impractical to deliver within HCs. However, cluster randomisation does carry a risk of contamination between clusters.[44] To minimise this risk, we have opted to stratify our clusters into zones proximate to and more distant from HCs, and only sample from the HC proximate zones. Nonetheless, we do acknowledge this more limited sampling frame may limit representativeness.

It must be remembered that primary care in Malawi is a system under stress.[17] Although HC-level records are used, these might be in electronic or paper-based forms, the latter being vulnerable to illegibility, damage and loss.[45] Thus, to further understand primary care antibiotic prescribing, a robust study method is needed which minimises disruption for already overstretched HC staff, while also enabling rapid informed consenting and data collection. For this reason, we sought to establish and optimise an ethical approach of verbally consented rapid HP review on exiting HCs.

Though not yet formally quantified, antibiotics are frequently informally (eg, private pharmacies and the local market) acquired in Malawi.[46] Thus, patient-held health records likely only represent a partial picture of a patient's disease history and antibiotic exposure.[16 45] For the community-based component of this study, we will implement visual recall methods previously utilised in this setting.[16] However, it should be remembered that we will remain reliant on participant recall. Similarly, while incidence of malaria would be the optimal endpoint, due to uncertainties surrounding access to diagnostic services, we are using febrile illness as a proxy. However, where recorded, we will also consider mRDT use and findings. To assist with this aim for both antibiotic exposure and febrile illness, we hope to also gain access to patient-held health records. This will enable comparisons to be made which will at least partially negate these limitations.

Considering further limitations, due to resource constraints the surveys will be cross-sectional, meaning that we will not be able to gain detailed understanding of seasonal variation in prescribing practices nor AMR. We will be able to manually summarise longitudinal prescription and mRDT use from largely paper-based HC-level health records; however, this will be resource intensive and does not represent a sustainable long-term approach. Finally, in the likely absence of reliable population data, we are reliant on using proportional outcome measures, though census data will be used where possible. It is possible that such proportional measures may mask wider

variation between intervention arms, for example, highly positive vaccine effects leading to absolute reductions in disease incidence.

To conclude, we present a protocol for a robust, pragmatic evaluation of pneumococcal and malaria vaccine impact on antimicrobial exposure, febrile illness and AMR carriage in young children, which considers the structural challenges of conducting such studies in a low-income country. Limitations considered, we are confident that this will provide a blueprint for wider evaluations to be conducted in other age groups and countries.

**Author affiliations**
[1]Department of Clinical Infection, Microbiology & Immunology, Institute of Infection, Veterinary and Ecological Sciences, University of Liverpool, Liverpool, UK
[2]Malawi-Liverpool-Wellcome Trust Clinical Research Programme, Kamuzu University of Health Sciences, Blantyre, Malawi
[3]Julius Center for Health Sciences and Primary Care, University Medical Centre Utrecht, Utrecht, The Netherlands
[4]Research Department of Infection, Division of Infection and Immunity, UCL, London, UK
[5]Department of Public Health, Kamuzu University of Health Sciences, Blantyre, Malawi
[6]Ministry of Health, Blantyre, Malawi
[7]Mangochi District Council, Mangochi, Malawi
[8]Ministry of Health, Lilongwe, Malawi

**Acknowledgements** The authors thank the MLW laboratory management team, led by Brigitte Denis and George Selemani. We also thank the MLW data management team, and would especially like to commemorate the memory of Clemens Masesa who led this team until his passing. We are further appreciative of the guidance and advice provided by Andrea Gori, Eleanor MacPherson, Priyanka Patel, Derek Cocker and Nicholas Feasey. We would also like to thank a range of supporting team members at MLW and KUHeS, particularly Mernani Kaonga and our data collection teams, in addition to the many HC team members, HSAs, health volunteers, community leaders and community members who without their kind support and time this study would not be possible.

**Contributors** RSH, NF, KMM, TDS and JeC conceived the study. DS, AI-P, RSH, NF, KMM, TDS, JeC and AK designed the study, with contributions from FB, NN, CG, RC, AK and WN. AI-P, TDS, JeC, CB and DS oversaw development of laboratory methods. DS, AI-P, FB and JeC designed study documents. JaC conducted community mapping exercises. GK, HC and CM provided public health oversight and facilitated access to study sites. DS and AI-P jointly wrote the first draft. All authors have read and approved the final manuscript.

**Funding** This work was funded by the Wellcome Trust (Ref: 219900/Z/19/Z) to RSH. The MLW Research Programme is supported by a Strategic Award from the Wellcome Trust, UK (206545/Z/17/Z). RSH is a National Institute of Health Research (NIHR) Senior Investigator.

**Disclaimer** The views expressed in this publication are those of the authors and not necessarily those of the NIHR or the UK Department of Health and Social Care.

**Map disclaimer** The inclusion of any map (including the depiction of any boundaries therein), or of any geographic or locational reference, does not imply the expression of any opinion whatsoever on the part of BMJ concerning the legal status of any country, territory, jurisdiction or area or of its authorities. Any such expression remains solely that of the relevant source and is not endorsed by BMJ. Maps are provided without any warranty of any kind, either express or implied.

**Competing interests** None declared.

**Patient and public involvement** Patients and/or the public were involved in the design, or conduct, or reporting, or dissemination plans of this research. Refer to the Methods section for further details.

**Patient consent for publication** Not applicable.

**Provenance and peer review** Not commissioned; externally peer reviewed.

**ORCID iDs**
David Singleton http://orcid.org/0000-0002-1980-5410
Todd D Swarthout http://orcid.org/0000-0001-5285-7039
Robert S Heyderman http://orcid.org/0000-0003-4573-449X

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
