## [Reviewer comments · BMJ Open]

ARTICLE DETAILS

TITLE (PROVISIONAL)	Cross-sectional health centre and community based evaluation of the impact of pneumococcal and malaria vaccination on antibiotic prescription and usage, febrile illness and antimicrobial resistance in young children in Malawi: the IVAR study protocol
AUTHORS	Singleton, David; Ibarz-Pavon, Ana; Swarthout, Todd; Bonomali, Farouk; Cornick, Jennifer; Kalizang'oma, Akuzike; Ntiza, Noah; Brown, Comfort; Chipatala, Raphael; Nyangulu, Wongani; Chirombo, James; Kawalazira, Gift; Chibowa, Henry; Mwansambo, Charles; Maleta, Kenneth; French, Neil; Heyderman, Robert

VERSION 1 – REVIEW

REVIEWER	khan, maria Rehman Medical Institute
REVIEW RETURNED	15-Feb-2023

GENERAL COMMENTS	Very good effort by the authors.
----------------------------------

REVIEWER	Palmu, Arto A. Natl Inst Hlth
REVIEW RETURNED	17-Feb-2023

GENERAL COMMENTS	Thank you for the opportunity to review this interesting protocol with valid design utilizing randomized studies with supplemental data collection. In general, this protocol deals with an important and relevant topic and I look forward to reading the eventual results of the study. The language is clear and well written. However, there is some room for better clarity in the text and figures. DETAILED REVIEWER COMMENTS The BMJ Open instructions for the protocol articles do not require a section of "Discussion" which is a disadvantage for an interested reader. Therefore, the strengths and weaknesses are only shortly reported in the "Article Summary". This should be improved by more clearly describing the rationale for the decisions on the protocol and mentioning the strengths and weaknesses of the protocol elsewhere in the text as the topic is described. Examples on this are missing details like  - estimated frequency of over-the-counter antibiotic use outside health care setting - coverage of the full malaria vaccination schedule in the study population - methods for blinding the study assessments - adjustment between the rural vs urban HCs for the malaria study The 2nd bullet point in the "Article Summary" is too general and not
--

	understandable for a reader. The chapter “Study site selection” should follow immediately the study setting for clarity. Further, 3+3 HCs are selected but the Figure 2 includes 4+3 HCs. Similarly, the text says “For IVAR, RTS,S/AS01-exposed (n=5 HCs) and non-exposed (n=3 HCs) clusters have been selected,” but the Figure 3 shows 3+3 HCs. row 218: b) Inclusion and exclusion criteria. It remains unclear whether repeated visits of the children previously enrolled are allowed.
--	---

REVIEWER	Hasso-Agopsowicz, Mateusz World Health Organization Department of Immunization Vaccines and Biologicals, Immunization Vaccines and Biologicals
REVIEW RETURNED	02-Mar-2023

GENERAL COMMENTS	The protocol describes a study that aims to measure the impact of two interventions on various AMR indicators: 1) switching from a 3+0 to a 2+1 schedule for a pneumococcal conjugate vaccine, and 2) introducing the malaria RTS,S vaccine. The protocol is well written and adequately describes the rationale and methodology for the study. However, I suggest that the protocol could be further strengthened by: 1) In the introduction, the authors should explain the scientific rationale behind the switch to the 2+1 schedule and how it would impact AMR. 2) The authors should explain why they chose the incidence of febrile illness as the endpoint instead of the incidence of malaria. 3) To measure the impact of the RTS,S vaccine, the authors could consider endpoints that would indicate resistance of the malaria parasite, such as the length of treatment or incidence of re-infection. 4) The authors propose measuring antibiotic prescription incidence as a proportion of outpatient department visits. However, it is possible that the vaccine could significantly improve a child's health and reduce the total number of visits, resulting in only more severe cases being seen at the clinic. In such cases, the proposed indicator would increase despite the vaccine's positive impact. The authors could consider other denominators, such as the number of children in the catchment area.
---

VERSION 1 – AUTHOR RESPONSE

Reviewer: 1
 Dr. Maria Khan, Rehman Medical Institute
 Comments to the Author:
 Very good effort by the authors.

Reviewer: 2
 Dr. Arto A. Palmu, Natl Inst Hlth
 Comments to the Author:
 Thank you for the opportunity to review this interesting protocol with valid design utilizing randomized studies with supplemental data collection. In general, this protocol deals with an important and relevant topic and I look forward to reading the eventual results of the study. The language is clear and well written. However, there is some room for better clarity in the text and figures.

DETAILED REVIEWER COMMENTS

The BMJ Open instructions for the protocol articles do not require a section of “Discussion” which is a disadvantage for an interested reader. Therefore, the strengths and weaknesses are only shortly reported in the “Article Summary”.

This should be improved by more clearly describing the rationale for the decisions on the protocol and mentioning the strengths and weaknesses of the protocol elsewhere in the text as the topic is described.

Examples on this are missing details like

- estimated frequency of over-the-counter antibiotic use outside health care setting
- coverage of the full malaria vaccination schedule in the study population
- methods for blinding the study assessments
- adjustment between the rural vs urban HCs for the malaria study

Thank you for this excellent suggestion. We have included a discussion reflecting on the points you have highlighted above (lines 490-548). Please note that to our knowledge there are no estimates of over-the-counter antibiotic use in Malawi to date – we have noted this in the discussion. Similarly, although vaccine coverage is something that we are collecting as part of this study, vaccine coverage rates for the malaria vaccine (RTS,S/AS01) in the general study population have not yet been publicly reported.

The 2nd bullet point in the “Article Summary” is too general and not understandable for a reader.

We have made this point a bit more specific (lines 41-43).

The chapter “Study site selection” should follow immediately the study setting for clarity. Further, 3+3 HCs are selected but the Figure 2 includes 4+3 HCs. Similarly, the text says “For IVAR, RTS,S/AS01-exposed (n=5 HCs) and non-exposed (n=3 HCs) clusters have been selected,” but the Figure 3 shows 3+3 HCs. row 218: b) Inclusion and exclusion criteria. It remains unclear whether repeated visits of the children previously enrolled are allowed.

We have moved the study site selection section earlier in the manuscript. Many thanks for spotting the omission on the RTS,S/AS01 evaluation, this was intended to refer to the 8 HCs already selected for the existing evaluation of which IVAR has selected 6. We have improved clarity in lines 158-160. Regarding repeated visits, we have clarified this in lines 235-236 and 338-339.

Reviewer: 3

Dr. Mateusz Hasso-Agopsowicz, World Health Organization Department of Immunization Vaccines and Biologicals

Comments to the Author:

The protocol describes a study that aims to measure the impact of two interventions on various AMR indicators: 1) switching from a 3+0 to a 2+1 schedule for a pneumococcal conjugate vaccine, and 2) introducing the malaria RTS,S vaccine. The protocol is well written and adequately describes the rationale and methodology for the study. However, I suggest that the protocol could be further strengthened by:

- 1) In the introduction, the authors should explain the scientific rationale behind the switch to the 2+1 schedule and how it would impact AMR.

We have included a brief rationale on lines 100-103.

2) The authors should explain why they chose the incidence of febrile illness as the endpoint instead of the incidence of malaria.

We have included a discussion of this in lines 527-532. We have also added lines 409-411 indicating that we will consider mRDTs as secondary outcomes.

3) To measure the impact of the RTS,S vaccine, the authors could consider endpoints that would indicate resistance of the malaria parasite, such as the length of treatment or incidence of re-infection.

From the data we are collecting we would indeed be able to consider these as additional outcomes and have included a note to this effect on lines 409-411. Due to issues with case follow-up, defining re-infection in this population may prove challenging. As such, we have opted to consider multiple positive mRDT results instead as a proxy. Length of prescribed antibiotic treatment has also been added as an outcome (line 404).

4) The authors propose measuring antibiotic prescription incidence as a proportion of outpatient department visits. However, it is possible that the vaccine could significantly improve a child's health and reduce the total number of visits, resulting in only more severe cases being seen at the clinic. In such cases, the proposed indicator would increase despite the vaccine's positive impact. The authors could consider other denominators, such as the number of children in the catchment area.

The Reviewer raises a difficult and important area. Unfortunately, there is an inconsistent availability of census data throughout much of Malawi, and as such in many cases other denominators are simply not available. This remains a limitation of this study. As suggested by Reviewer 2, we have now included a discussion which addresses this point (lines 539-542). However, we have also included the possibility of taking the approach suggested by yourself as an additional outcome (lines 247-249).

VERSION 2 – REVIEW

REVIEWER	Palmu, Arto A. Natl Inst Hlth
REVIEW RETURNED	25-Apr-2023
GENERAL COMMENTS	The authors' responses are satisfactory. No further comments on the revised version.
REVIEWER	Hasso-Agopsowicz, Mateusz World Health Organization Department of Immunization Vaccines and Biologicals, Immunization Vaccines and Biologicals
REVIEW RETURNED	25-Apr-2023
GENERAL COMMENTS	I would like to thank the authors for incorporating the suggested changes. The paper is well-written and I recommend it for publication.